# Oxidative Stress and the Nuclear Factor Erythroid 2-Related Factor 2 (Nrf2) Pathway in Multiple Sclerosis: Focus on Certain Exogenous and Endogenous Nrf2 Activators and Therapeutic Plasma Exchange Modulation

**DOI:** 10.3390/ijms242417223

**Published:** 2023-12-07

**Authors:** Dimitar Tonev, Albena Momchilova

**Affiliations:** 1Department of Anesthesiology and Intensive Care, University Hospital “Tzaritza Yoanna—ISUL”, Medical University of Sofia, 1527 Sofia, Bulgaria; 2Institute of Biophysics and Biomedical Engineering, Bulgarian Academy of Science, 1113 Sofia, Bulgaria; albena_momchilova@abv.bg

**Keywords:** multiple sclerosis, oxidative stress modulation, Nuclear Factor Erythroid 2-Related Factor 2 pathway activation, exogenous, endogenous, nerve growth factor, therapeutic plasma exchange

## Abstract

The pathogenesis of multiple sclerosis (MS) suggests that, in genetically susceptible subjects, T lymphocytes undergo activation in the peripheral compartment, pass through the BBB, and cause damage in the CNS. They produce pro-inflammatory cytokines; induce cytotoxic activities in microglia and astrocytes with the accumulation of reactive oxygen species, reactive nitrogen species, and other highly reactive radicals; activate B cells and macrophages and stimulate the complement system. Inflammation and neurodegeneration are involved from the very beginning of the disease. They can both be affected by oxidative stress (OS) with different emphases depending on the time course of MS. Thus, OS initiates and supports inflammatory processes in the active phase, while in the chronic phase it supports neurodegenerative processes. A still unresolved issue in overcoming OS-induced lesions in MS is the insufficient endogenous activation of the Nuclear Factor Erythroid 2-Related Factor 2 (Nrf2) pathway, which under normal conditions plays an essential role in mitochondria protection, OS, neuroinflammation, and degeneration. Thus, the search for approaches aiming to elevate endogenous Nrf2 activation is capable of protecting the brain against oxidative damage. However, exogenous Nrf2 activators themselves are not without drawbacks, necessitating the search for new non-pharmacological therapeutic approaches to modulate OS. The purpose of the present review is to provide some relevant preclinical and clinical examples, focusing on certain exogenous and endogenous Nrf2 activators and the modulation of therapeutic plasma exchange (TPE). The increased plasma levels of nerve growth factor (NGF) in response to TPE treatment of MS patients suggest their antioxidant potential for endogenous Nrf2 enhancement via NGF/TrkA/PI3K/Akt and NGF/p75NTR/ceramide-PKCζ/CK2 signaling pathways.

## 1. Introduction

Multiple sclerosis (MS) is an autoimmune multifocal inflammatory disease of the central nervous system (CNS) characterized by chronic inflammation, demyelination, axonal damage, and subsequent gliosis. The CNS regions that are commonly affected by MS include the periventricular region, the subcortical region, the optic nerve, the spinal cord, the brain stem, and the cerebellum. According to the natural course of the disease, it is categorized as relapsing–remitting, secondary progressive, and primary progressive [1]. The etiology of MS presumes that, in genetically pre-disposed patients, T lymphocytes become activated in the peripheral compartment, pass through the blood–brain barrier (BBB), and cause tissue damage in the CNS compartment. They produce proinflammatory cytokines; induce cytotoxic functions in microglia and astrocytes with the accumulation of reactive oxygen species (ROS), reactive nitrogen species (RNS), and other superoxide radicals; stimulate B cells and macrophages; and activate the complement system [2]. The imbalance between ROS accumulation and clearance underlies oxidative stress (OS), and this plays a critical role in regulating cell and tissue physiological and pathological processes [3]. Inflammation and neurodegeneration are involved from the very beginning of the disease [4]. They can both be affected by OS with different emphases depending on the time course of MS. Thus, in the acute phase OS initiates inflammatory processes, while in the chronic phase, it supports neurodegeneration [5]. A still unresolved issue in overcoming OS-induced lesions in MS is the failure to activate the nuclear factor erythroid 2-related factor 2 (Nrf2) signaling pathway, which basically plays a critical role in preventing mitochondrial failure, OS, neuroinflammation, and degeneration [6]. The level of endogenous stimulation of the Nrf2 pathway is not sufficient to counteract OS overload. Thus, the application of exogenous approaches that are capable of elevating endogenous Nrf2 stimulation can contribute to protecting the brain tissue against oxidative damage [7]. However, exogenous Nrf2 activators themselves are not without drawbacks [8,9,10], necessitating the search for new non-pharmacological approaches to modulate OS. The purpose of this review is to present some relevant preclinical and clinical examples, focusing on certain exogenous and endogenous Nrf2 activators and the therapeutic plasma exchange (TPE) modulation of OS in MS.

In this focus review, along with a short summary of OS and the Nrf2 pathway in MS, we explore some relevant natural and synthetic compounds, nerve growth factor (NGF) and TPE in the context of their role in Nrf2 activation, and OS modulation.

## 2. Methodological Approaches

A search in the literature was performed through September 2023 of MEDLINE, EMBASE, and Cochrane Library based on the Medical Subject Heading (MeSH) of “oxidative stress”, “OS”, “reactive oxygen species”, “ROS”, “reactive nitrogen species”, “RNS”, “superoxide radicals”, “oxidants”, “antioxidants”, “enzymatic”, “non-enzymatic”, “balance”, “modulation”, “nuclear factor erythroid 2-related factor 2”, “Nrf2”, “Keap1/Nrf2/ARE”, “Nrf2/HO-1”, “signaling”, “pathway”, “activation”, “activators”, “exogenous”, “endogenous”, “natural”, “synthetic”, “pharmacological”, “non-pharmacological”, “multiple sclerosis”, “MS”, “acute”, “chronic”, “relapsing remitting”, “secondary progressive”, “primary progressive”, “aggressive”, “attacks”, “exacerbations”, “relapses”, “central nervous system”, “CNS”, “inflammation”, “neuro-inflammation”, “demyelination”, “degeneration”, “neuro-degeneration”, “neurodegenerative”, “diseases”, “neuroprotection”, “nerve growth factor”, “NGF”, “plasma levels”, “neurotrophins”, “receptors”, “tropomyosin receptor kinase A”, “TrkA”, “p75 neurotrophin receptor”, “p75NTR”, “therapeutic plasma exchange”, “nanomembrane-based”, “plasmapheresis”, and “apheresis”, as well as through a manual search in the local database (National Library of Saints Cyril and Methodius, Sofia, Bulgaria). The search had no language restrictions.

## 3. OS in MS

OS plays a key role in neurodegenerative processes by inducing oxidative damage to lipid, protein, and deoxyribonucleic acid (DNA) molecules. Oxidative destruction of proteins makes them alter their active configuration and form oligomers with different functions or molecular fragments that cause pro-inflammatory processes that aggravate OS. Mitochondrial DNA also undergoes alterations caused by ROS and RNS, leading to misfunctions in the key proteins involved in key cellular metabolic processes. The impairment of mitochondrial activity leads to the accumulation of ROS and activation of pro-apoptotic pathways. Lipid peroxidation caused by ROS destroys the structure, asymmetry, and permeability of cell membranes’ bilayer, leading to an enhanced inflammatory response, changes in calcium homeostasis, and neuronal death [11,12,13]. Moreover, the degree of oxidative attack towards proteins, lipids, and DNA frequently is sufficient to generate neoepitopes due to a loss of immunological tolerance, leading to the development of autoimmunity [14,15,16]. The latter implication increases the relevance of using TPE as a means of both modulating OS and ameliorating its adverse consequences.

OS is a well-recognized pathogenic factor in the onset and development of a vast variety of neurodegenerative pathologies, including MS. This neurodegenerative disease with a pronounced neuroinflammatory component is characterized by different symptoms and clinical manifestations, which are basically accompanied by inflammation-related damage to the CNS and different levels of disturbed motor activity in affected patients [17]. Inflammation elevates the content of free radicals, causing a higher number of oxidative stress biomarkers [18]. OS is generally an impaired oxidant/antioxidant balance that can cause neuroinflammation and neurodegeneration [19]. The imbalance is characterized by increases in ROS/RNS, the pro-inflammatory transcriptional factor, enzymatic oxidants, and oxidation end products at the expense of decreases in the antioxidant transcriptional factor and enzymatic and non-enzymatic antioxidants (Figure 1). Since OS may be associated with neurodegenerative disorders, an antioxidant-related therapeutic approach might be a promising perspective [20]. OS has also been assumed to be a significant pathogenic factor underlying the etiology of MS [21].

OS, mitochondrial damage, and energy are possibly implicated in plaque formation and neurological degeneration in lesions of white and grey matter [23,24,25]. The brain tissue is a target of oxidative attack because of the high oxygen requirements for its functional activity, significant amount of polyunsaturated fatty acids (PUFAs), and comparatively low activity and level of antioxidant enzyme systems [26]. ROS and RNS play an essential role in MS occurrence and development by contributing to the impairment of oligodendroglia functions [27]. According to the literature, the relapsing–remitting and progressive phenotypes of MS are determined by different mechanisms. For instance, focal inflammatory processes are associated with the development of relapses, while axonal degeneration may be associated with disease progression. OS in MS is presumed to be associated with both processes in MS patients [26,28]. ROS play a multifunctional role in the formation of MS lesions [29]. ROS are known to accumulate due to the interaction between monocytes and the brain endothelium, the latter leading to tight-junction modifications and disturbance of the blood–brain barrier’s integrity, which induces the migration of leukocytes into the CNS [29,30]. In addition, the infiltrated leukocytes induce an accumulation of higher amounts of ROS, which underlie myelin degradation [31], destruction of the oligodendroglia [32], and finally neuronal and axonal impairment [33].

The major sources of ROS include the mitochondria, peroxisomes, and phagocytic cells. Macrophages occur as an important source of ROS [34] due to high oxygen consumption [35,36]. Recent studies in experimental models of MS have reported that the rapid response of resident immune cells in the CNS, most notably the pro-inflammatory polarization of microglia [37] and astrocytes [38], is associated with the expression and release of reactive molecules associated with OS [39,40]. They have a significant effect on chronic myelin loss and inhibit the remyelination and repair of local CNS lesions [41]. Thus, activated microglia and astrocytes are often considered a major source of ROS and RNS in the CNS [42].

Myelin, which surrounds the axons and also acts as a major target of immune attacks in MS, consists of about 30% protein and 70% lipids. Malondialdehyde (MDA) is a marker of lipid peroxidation and is a result of PUFA peroxidation [18]. This aldehyde provides information on the level of OS by evaluating the oxidative damage of lipids [18]. Isoprostanes, which also serve as markers of lipid oxidative damage, are prostaglandin F-like products of the peroxidation of unsaturated acyl chains, predominantly of arachidonic acid [43,44]. The content of isoprostanes in plasma have shown to be elevated in MS, including in relapsing–remitting MS and secondary progressive MS types [45,46]. There are reports showing that isoprostanes and MDA are increased in the plasma of MS patients compared with controls and support their role as indicators of disease development, disability progression, or therapeutic response [47]. Along with altering ROS plasma levels, they reflect the success or failure of antioxidant treatment in MS patients [17].

## 4. Nrf2 Pathway in MS

Under homeostatic conditions, Nrf2 is localized in the cytosol through its negative regulator, kelch-like ECH-associated protein 1 (Keap1), and is subject to proteasomes disintegration through a couple of ubiquitin ligase systems [48,49]. Thus, the Keap1/Nrf2 interaction underlies a low expression of Nrf2-regulated genes. OS or Nrf2 activators, along with non-pharmacologic Nrf2 modulation, break the complex between Nrf2 and Keap1, inducing the translocation of Nrf2 towards the nucleus. There, Nrf2 heterodimerizes with small musculoaponeurotic fibrosarcoma (sMaf) proteins and binds to antioxidant response elements (ARE) in the promoter region of target genes [48,49,50,51,52]. This region encodes several antioxidant enzymes that neutralize ROS and electrophiles, involving superoxide dismutase (SOD), glutathione peroxidase (GPx), catalase, glutathione reductase, and heme oxygenase-1 (HO-1) (Figure 2) [53,54].

Autopsy specimens of MS patients show the upregulation of Nrf2 and the Nrf2-responsive genes HO-1 and NQO-1, in and near to the active lesions in spinal cord and brain samples [55,56], as an integral part of the cellular anti-oxidative response. The increased expression of Nrf2 has been observed in astrocytes and macrophages in active lesions [54] and in oligodendrocytes at the lesion’s edges [55]. However, the level of Nrf2 expression is cell-type-specific [57]. In the CNS, astrocytes harbor a more efficient anti-oxidative potential compared to neurons. The expression of Nrf2 is lower in neurons, even when they are surrounded by Nrf2-positive glia [56]. Consistent with the essential role of Nrf2 for ROS detoxification, Nrf2-deficient cells are more susceptible to oxidative destruction, etc. [58]. This may underlie a limited capacity of neurons to cope with OS.

In experimental autoimmune encephalomyelitis (EAE), an established model for MS, Nrf2-deficient mice show a more rapid onset, an exacerbated clinical severity, an increased number of lesions and infiltrating immune cells, greater microglial activation, and visual dysfunction [59,60,61]. Additionally, and importantly, the oligodendrocytes damaged in the EAE lesions have relatively low levels of Nrf2. Therefore, low levels of Nrf2 or the impaired activation of Nrf2 in oligodendrocytes may account for selective susceptibility in neuroinflammatory conditions due to the high vulnerability of these oligodendrocytes to OS [62].

Dysfunctions in the Nrf2 signaling pathway result in the impairment of redox homeostasis, which leads to ROS/RNS overload [63] and the prevalence of other redox-sensitive transcription factors such as activator protein 1 and the pro-inflammatory nuclear factor kappa-light chain-enhancer of activated B cells (NF-kB). The latter stimulates the expression of certain genes involved in MS pathogenesis, such as tumor necrosis factor α (TNF-α), iNOS, interleukin 1α/β (IL-1α/β), and some growth factors [64]. There is a crosstalk between the Nrf2 and NF-ĸB pathways. Nrf2 binds with its transcriptional cofactor cAMP-response-element-binding protein (CBP) to initiate ARE-driven gene expression. When NF-ĸB binds with CBP in a competitive way, it hinders the binding between CBP and Nrf2, the latter leading to inhibition of Nrf2 activation. Thus, NF-kB and Nrf2 compete for CBP, which promotes DNA binding [65]. Given that OS and inflammation are closely related processes in MS, the activation of the Nrf2 pathway interferes with both processes in a complex manner [66].

HO-1, one of the most important antioxidant enzymes downstream of Nrf2, plays an important role for the anti-inflammatory activity in EAE and MS [67]. Decreased expression of HO-1 was found in peripheral blood mononuclear cells (PBMCs) of MS patients, and a significant downregulation of this enzyme was observed during disease exacerbations [68]. These findings indicate a higher likelihood of relapse in patients with reduced HO-1 expression in PBMC and are confirmed by the results of a microarray meta-analysis [68]. In this context, HO-1 inducers may occur as important factors in the treatment of MS.

A thorough understanding of Nrf2-mediated HO-1 gene transactivation requires taking into account the cooperation or competition with other transcriptional factors at ARE and ARE-like sites [69]. The activating transcription factor 4 (ATF4) can dimerize with Nrf2 at ARE to promote the expression of HO-1, whereas the transcription repressor BTB and CNC homology 1 (BACH1), which are critical in the regulation of the HO-1 gene, compete with Nrf2 at the ARE binding sites, and Nrf2-induced expression requires the inactivation of BACH1 sites [69]. In addition, the expression of HO-1 is downregulated by BACH1 when the heme content is low, but higher heme levels inhibit BACH1–DNA binding and promote BACH1 exportation and degradation [70]. TPE techniques using peripheral venous access and high transmembrane pressure cause hemolysis during the procedure [71], which could contribute to BACH1 inhibition as well. Likewise, the significant decrease in plasma proteins and other plasma constituents observed during TPE [72] could contribute to amino acid losses, a well-known inducer of ATF4 [73]. Thus, TPE could modulate HO-1 expression indirectly at multiple levels.

In summary, OS and antioxidant Nrf2 pathways are important players in the pathophysiology of MS and represent a promising target for approved or investigational pharmacological and non-pharmacological therapies of MS [57].

## 5. The Role of Certain Natural and Synthetic Compounds as Exogenous Nrf2 Activators

The redox homeostasis in the brain tissue is under the control of Nrf2, and there has been a growing interest in identifying natural or synthetic compounds (Figure 3) that are able to modify Nrf2 activity, for example, in MS, in which ROS/RNS have a recognized function [74].

Curcumin (Figure 3A) has been tested as a component of the therapeutic protocol of MS [75,76,77]. Its immunomodulatory and anti-inflammatory properties are mainly mediated by its capacity to inhibit COX-2, iNOS, and also the NF-kB transcription pathway [78]. It reduces the clinical severity of EAE, the content of inflammatory cells infiltrating the spinal cord, the production of IL-17, and the generation of TNF-α, accompanied by the upregulation of Treg cells, suggesting that curcumin enhances Th2/Treg responses in EAE [79,80]. Only one study revealed the role of curcumin in increasing the expression of Nrf2/HO-1 in an EAE model of MS [81]. However, some significant drawbacks, such as poor solubility, low bioavailability, and dose-dependent pro-oxidant activity, hinder its wider clinical use.

Resveratrol (Figure 3A) affects various targets through different pathways, most notably Nrf2, silent information regulator T1 (SIRT1), and NF-κB [66]. In preclinical animal studies, it decreases the disease severity [82,83], the inflammatory infiltrates [84], and the NF-kB signaling [85] and improves the antioxidant potential through Nrf2 activation [85]. However, one study reveals the exacerbation of disease severity. The authors suggest that the worsening of EAE may be a result of the increased migration of inflammatory cells across the BBB, since resveratrol acts as a vasodilator, resulting in improved endothelial cell function, increased blood flow volume, and decreased blood flow velocity [86]. Like curcumin, resveratrol has poor bioavailability. Few clinical studies have evaluated the efficacy of resveratrol as a component of the therapeutic protocol of neurodegenerative diseases (Alzheimer’s disease and Friedreich Ataxia) [87,88], but none have addressed its potential in MS.

Epigallocatechin gallate (EGCG) (Figure 3A) is the most investigated flavonoid that occurs that has the greatest capacity for improving MS and EAE. There are studies showing beneficial effects of this flavonoid related to the reduced migration of immune cells beyond the BBB, reduced demyelination in the spinal cord, and decreased secretion of pro-inflammatory cytokines [89,90,91,92,93,94,95]. No clinical studies have addressed the significance of Nrf2 as a mechanism of anti-inflammatory and antioxidant activity of EGCG. However, EGCG has been shown to activate Nrf2 in animal models of Alzheimer’s disease [96] and ischemic stroke [97]. The results of clinical trials investigating the neuroprotective potential of EGCG are inconclusive [66].

Sulforaphane (Figure 3A) is a potent Nrf2 inducer [98]. It is 13.5 times more potent than curcumin and 105 times more potent compared to resveratrol, as evaluated through the induction of NQO-1 activity [99], with much better tissue penetration, absorption, and bioavailability compared to curcumin and resveratrol [51]. Studies on the mechanism of action of sulforaphane have revealed that it enhances the expression of HO-1 and NQO1 via upregulation of the Nrf2 pathway, thereby reducing the levels of OS. The direct effect of sulforaphane on microglia is manifested by increasing Nrf2 DNA-binding activity and the expression of Nrf2 downstream genes (NQO-1 and HO-1) [100]. It also reduces T-cell-mediated immunity and inflammation. Moreover, it reduces demyelination and CNS infiltration and inhibits the production of metalloproteinase 9 (related to the degradation of the extracellular matrix), which contributes to the preservation of the BBB [95]. Thus, it inhibits the development of EAE in mice through its antioxidant and anti-autoimmune inflammatory properties [101,102].

Melatonin (Figure 3A) acts as antioxidant and immunomodulatory agent, with either pro- or anti-inflammatory functions in a context-dependent way [103]. It offers potent protection against the DNA oxidative damage through a variety of mechanisms, including an increase in antioxidant enzymes [104]. The activation of antioxidant enzymes results predominantly from the stimulation of Nrf2 and its downstream proteins HO-1, NQO-1, and SOD [104,105,106,107]. The majority of preclinical studies show the beneficial effect of melatonin in treating EAE, but only one study shows its association with the upregulation of the Nrf2/HO-1 pathway [108]. In a small clinical trial, melatonin supplementation led to a statistically significant increase in SOD and GPx in the erythrocytes of secondary progressive MS patients. There was a positive correlation between SOD levels and the Expanded Disability Status Scale (EDSS) score both before and after melatonin treatment [109], indicating the importance of antioxidant protection in controlling disability in MS.

Oleanolic acid (CDDO) has been used as a source for the synthesis of several novel synthetic derivatives with improved neuroprotective properties due to Nrf2 activation. Among them, ethyl amide (CDDO-EA) (Figure 3B), methyl-amide (CDDO-ME or Bardoloxene Methyl) (Figure 3B), and trifluoroethyl amide (CDDO-TFEA) (Figure 3B) have shown the capacity to attenuate the clinical severity of EAE models, decrease Th1 and Th17 cytokines, diminish expression of iNOS in basal conditions, and upregulate HO-1 expression (over 3-fold relative to control conditions), which is consistent with the Nrf2 pathway upregulation [110,111].

Dimethyl fumarate (DMF) (Figure 3B) is a prodrug that is converted to monomethyl fumarate (MMF) [112], which is able to induce Nrf2 in different models, such as in astrocytes (over 15-fold increase in HO-1-mRNA levels at 10 μM) [113]. The administration of DMF resulted in the increased expression of NQO-1 and HO-1 genes in ex-vivo-stimulated peripheral blood mononuclear cells (PBMCs) as well [114]. Its mechanism of action is related to the direct modification of the Nrf2 inhibitor, Keap-1 [61], activating Nrf2 and thereby protecting neurons and oligodendrocytes from oxidative damage [61], as well as to its ability to increase glutathione (GSH) levels to inhibit NF-κB translocation, thereby reducing the expression of NF-κB-dependent genes (proinflammatory cytokines, chemokines, and adhesion molecules) [115]. DMF inhibits leukocyte migration and thereby the infiltration of immune cells in the CNS [116,117]. It could also directly mediate a shift toward Th2 instead of Th1 differentiation [118]. In the EAE model, DMF is reported to reduce symptom severity and preserve myelin and axon density [61]. Clinical trials have demonstrated that DMF significantly reduces the number of lesions [119], the annualized relapse rate, the risk of confirmed disability progression, the new or enlarging hypertense lesions, and the gadolinium-enhanced lesions [120,121]. Since 2013, DMF has been approved by the FDA and the EMA as a first-line therapy for relapsing–remitting MS [122].

However, DMF has relatively low efficacy and specificity [10]. The clinical use of DMF has been associated with the occurrence of a number of adverse effects, such as abdominal pain, nausea, and treatment-related progressive multifocal leukoencephalopathy along with long-lasting, severe lymphocytopenia [112,123,124,125]. In a clinical trial of DMF in patients with relapsing–remitting MS, the induction of the NRF2 transcriptional target NQO1 was found to negatively correlate with the age of the patient [126]. All these disadvantages of DMF may shift the focus to the use of other approved drugs or new non-pharmacological therapeutic approaches with similar Nrf2-inducing potential. Fingolimod (a classical S1PR modulator) regulates mitochondrial OS in neuronal cells by enhancing Nrf2 activity with increased HO-1 and NQO1 gene expression [127]. Siponimod (a novel S1PR modulator) activates Nrf2 while hampering NFκB in human astrocytes, suggesting direct anti-inflammatory and antioxidant pharmacological effects [128] (Figure 4). Another clinically approved drug that has also been reported to activate Nrf2, together with other anti-inflammatory mechanisms of action, is natalizumab, an antibody targeting the cell adhesion molecule α-4 integrin [129]. The role of TPE in the context of Nrf-2 activation will be discussed below.

## 6. The Role of NGF as an Endogenous Nrf2 Activator

NGF is the most recognized neurotrophin, and it was described in 1952 by Levi-Montalcini [130]. NGF activates two types of membrane receptors, tropomyosin receptor kinase A (TrkA) and p75 neurotrophin receptor (p75NTR) [131]. Depending on the NGF-specific receptors, signaling pathways associated with neuronal differentiation, maturation and survival, axonal and dendrite development, or apoptosis can be stimulated. When TrkA and p75NTR are co-expressed, they comprise a two-receptor heterotetrameric complex that binds to NGF, resulting in the activation of various signaling pathways [132,133].

In neurological disorders with neuroinflammation, almost all resident cells of the CNS overexpress NGF [134]. Moreover, NGF can cross the BBB when the BBB becomes permeable in pathological conditions, such as MS [135]. It is noteworthy that NGF levels influence glial physiology. There is evidence that in an in vivo mouse model [136], the depletion of NGF causes pro-inflammatory astrocyte activation and neurotoxicity. Conversely, the upregulation of NGF directs microglia to an anti-inflammatory phenotype [137], thereby leading to neuroprotection. The latter is of particular importance in the context of OS in MS, considering the role of pro-inflammatory microglia and astrocytes in the generation of ROS/RNS.

Previous studies have observed the significant effects of the intracranial administration of NGF on the downregulation of cytokines (important factors in OS) that are specific to the CNS parenchyma and not detected in the periphery [138]. In the case of MS, during acute attacks, patients show increased levels of NGF in CSF compared to healthy individuals, which can be seen as an attempt to protect the CNS tissue against inflammation [139]. There, observations imply the relevance of NGF-related therapeutic schemes in patients with inflammatory and neurodegenerative pathologies [136]. Last but not least, NGF antibodies have been observed to exacerbate the neuropathological signs of EAE [140]. This suggests not only the importance of NGF in decreasing the EAE lesions [141], but also offers new possibilities for increasing the anti-inflammatory potential of NGF in MS patients by removing these antibodies using TPE [142,143].

Besides the already-mentioned anti-inflammatory potential, NGF plays an important role in antioxidant protection in experimental and clinical settings. As early as the 1990s, preclinical studies outlined the role of NGF in reducing OS [144,145]. The activation of the MAPK pathway appears to be required for this action of NGF (namely acute suppression of neuronal ROS production by NGF), implicating TrkA signaling in defense against oxidative damage [145,146]. Other research from the late 1990s and the early next decade revealed the role of NGF in the upregulation of enzymatic antioxidants (SOD, HO-1, GPx, and catalase) through phosphoinositide-3-kinase (PI3K)/Akt and NF-kB signaling [22,147,148,149]. More recent studies have shown that the activation of TrkA by NGF induces the nuclear translocation of Nrf2 and subsequently induces the transcription of ARE-containing genes [150,151,152]. Moreover, the activated Nrf2 can in turn up-regulate NGF gene expression and a positive feedback loop between Nrf2 and NGF can be formed [150,151], which further strengthens the evidence for the potential of NGF as an Nrf2-inducer. New evidence suggests that NGF is able to activate p75NTR/ceramide-protein kinase C-ζ (PKCζ)/casein kinase 2 (CK2) signaling pathways through direct interaction with p75NTR in order to mediate its activation of Nrf2 [153] (Figure 5).

Most interesting in view of future clinical benefits is a recent preclinical study whose results provide evidence that the depletion of peripheral GSH pools increases peripheral circulating NGF, which orchestrates a neuroprotective response in the CNS, at least in the striatum, via the NGF/TrkA/Akt/Nrf2 pathway [154]. A plausible explanation for the role of NGF after systemic OS suggests an increased synthesis of NGF in peripheral tissues, followed by secretion and increased levels in the systemic circulation. Peripheral NGF then reaches the cerebral circulation and activates the TrkA pathway in brain endothelial cells, which induces the transcription of sulfhydryl AAs L-cys/L-cys2 transporters in neurons and glial cells. Given that brain GSH synthesis is limited by the presence of the sulfhydryl AAs L-cys and L-cys2 [155], it has been suggested that the activation of the NGF/TrkA/Akt/Nrf2 signaling pathway in the striatum may be related to the transcriptional regulation of amino acid transporter genes associated with the presence of L-cys and L-cys2 in the brain. In addition, NGF synthesis is elevated in brain endothelial cells, and NGF is secreted into the brain parenchyma, activating the NGF/TrkA/PI3K/Akt/Nrf2 pathway, which stimulates the expression of antioxidant genes in the CNS. Another possibility is that peripheral NGF enters the CNS directly, as suggested in the 1980s by Levi-Montalcini [156], which may occur with a permeable BBB, as is the case in acute exacerbations of MS [135,143]. Translating the results of this preclinical model to the real clinical setting is achievable through the application of TPE in the treatment of acute exacerbations of MS. Through this non-pharmacological therapeutic approach, both an increase in circulating levels of NGF and a modulation of OS can be achieved [17,143]. Evidence for this will be presented in the next section.

## 7. The Role of TPE as an OS Modulator

TPE is a therapeutic approach that involves extracorporeal blood removal, as well as an exchange of blood plasma or components. It basically removes a significant volume of plasma (1 to 1.5 of the patient’s total plasma volume (TPV) per treatment) which is accompanied with corresponding volume replacement using colloid solutions or a combination of crystalloid/colloid solutions [157]. The aim of TPE is to remove potentially pathogenic substances with high molecular weight (>150 kDA), involving autoantibodies, pro-inflammatory mediators, lipids, and other molecules or molecular fragments from the intravascular space [157,158]. However, the mechanism of TPE action in inflammatory and neurodegenerative disorders involves more than just a removal of potentially a number of pathogenic molecules. It should be noted that TPE may also modulate cellular immunological response by changing the ratio between T helper type-1 (Th-1) and type-2 (Th-2) cells. Th-2 cells regulate the humoral immune response by facilitating B-cell autoantibody production and play an important role in the development of neurodegenerative autoimmune pathologies. By shifting the balance between peripheral T cells from Th-2 to Th-1 predominance, TPE modulates the pathological immune response and thus might induce a beneficial effect [159].

Our experience in the TPE treatment of MS exclusively involves the use of nanomembrane-based TPE (Figure 6) [17]. This innovative method involves passage of the patient’s blood through nanomembranes designed to filter out definite molecules [18,142,160,161,162,163,164]. There are pores in the nanomembrane with a diameter of 30–50 nm, which allow the filtration of molecules with a molecular weight below 40 kDa. The device has a volume of up to 70 mL [165]. The most widely used replacement is saline (NaCl 0.9), which is cheap and has no adverse effects, even when 25% of the circulating plasma is replaced [162]. The use of saline to replace the removed plasma is in agreement with the so-called low-volume plasma exchange (LVPE), which ranges from 350 mL to 2 l plasma volume removal for each separate procedure [166,167,168,169].

We performed a nanomembrane-based TPE treatment in steroid-refractory MS in five patients with a relapsing–remitting MS [18,143,164]. Most patients received three TPE procedures in LVPE mode (patients 1, 2, 3, 5), whereas one of them received four nanomembrane-based TPE procedures in LVPE mode (patient 4). We observed an elevation in NGF plasma levels in all patients (Figure 7) [143]. The measurement of OS markers (ROS and isoprostanes) showed that the application of TPE leads to a reduction in the OS level (Figure 8 and Figure 9) [17], which is favorable for the general and neurological status of patients.

It has been found that TPEs, mainly in patients with a lower degree of disability (Kurtzke EDSS ≤ 5 points for the RR form of MS [19], i.e., with predominant inflammation in the relapse occurring, not neurodegenerative), have a relatively favorable effect on clinical symptoms, which leads to an overall stabilization in the course of the disease. A significant decrease of 0.5 points on Kurtzke EDSS was assumed to establish the responder rate [168]. Only one patient (patient 3) had no improvement in the evaluated clinical disability (Kurtzke EDSS before TPE 8.5 points, after TPE 8.5 points). In these cases, also related to measurement of the OS before and after TPE [18,164], an improvement (according to Kurtzke EDSS, Table 1 [17]) of about 80% was found, which was consistent with the 74% improvement in the same type of patients (the relapsing–remitting form of MS) published by other authors [170].

TPE is presumed to affect NGF in many different ways. The most probable explanation for the reported increase in NGF levels in the plasma of our MS patients could be related to the removal of autoantibodies against NGF-producing cells and NGF itself [143]. The elevated NGF content after TPE could either contribute to or occur as a consequence of augmented NGF levels in the CNS (considering the NGF’s ability to pass the BBB in accordance with its gradient) [135]. Whatever the explanation, the observed results can be considered as a beneficial anti-inflammatory effect induced by the decreased inflammatory level (autoantibodies, immune complexes, cytokines, etc.) related to discarded plasma.

In addition, we observed a decrease in the plasma levels of sphingosine-1-phosphate (S1P) due to its direct loss with the discarded plasma as well [143]. It is believed that S1P has antioxidant effects that occur by potentiating the antioxidant enzymes SOD and catalase [171]. Thus, by reducing the S1P-like systemic GSH depletion mentioned above, we may postulate another possible mechanism in order to explain the elevated circulatory levels of NGF found by others in preclinical settings [154]. Long before us, however, other authors investigated the parameters of OS after low-density lipid apheresis and found an increase in plasma GSH levels [172]. While GSH can be removed via hemodialysis due to its low molecular weight, this is clearly not valid for low-density lipid apheresis [172]. The particular disease for which the corresponding blood purification technique is applied really matters.

There is another factor that may influence OS during TPE treatment in a context-dependent manner. In the field of neuroscience, the level of plasma albumin is mainly studied as a measure of OS at different levels of disease activity [173]. Consistent with this, a recent meta-analysis identified hypoalbuminemia in MS as strong clinical evidence of increased OS [173]. However, there are controversial data on cohorts without hypoalbuminemia in MS patients as well [174,175]. So far, there is only one study that longitudinally analyzed the effect of plasmapheresis or immunoabsorption (IA) on albumin quantity and redox state in patients with autoimmune-mediated neurological disorders [176]. The study showed that plasmapheresis induced a severe and long-lasting change in endogenous irreversibly oxidized albumin levels during the course of treatment and 12 days after the last plasmapheresis, which was not the case in patients treated with IA [176]. Nevertheless, they found a mean decrease in albumin concentration of about 4.7 g/l during one IA treatment with the proportion of increased irreversibly oxidized albumin over five IA treatments within the normal range [176]. It should be noted that IA is an apheresis technique where commercial albumin is not administered as a replacement fluid (just as is the case with our nanomembrane-based TPE). Interestingly, in our five MS patients presented above (all with reduced OS, Figure 7 and Figure 8), we found almost the same mean decrease in albumin concentration of about 4.6 g/l during a single treatment with nanomembrane TPE (Figure 10) [177]. Different apheresis techniques probably matter. As for the difficult question of to what extent the technological improvement in the apheresis membrane could improve outcomes, only future large-scale studies could provide an evidence-based answer.

## 8. Summary and Conclusions

Redox homeostasis in the brain is controlled by Nrf2, and we have witnessed increasing interest in identifying natural or synthetic compounds to modify Nrf2 activity [74]. However, their application is associated with some significant limitations, such as poor solubility, low bioavailability, and dose-dependent pro-oxidant activity [66]. Basically, Nrf2 activation plays a beneficial role under physiological conditions, but it promotes cancer development and metastasis, and also anticancer drug resistance after cancer is established [3]. In addition, exogenous antioxidants are also susceptible to oxidation, and therefore, their use as foods or supplements should be carefully considered [178]. The administration of high doses of antioxidants is considered to potentially expose the body to additional risks, as there is increasing evidence of some harmful effects, such as the risk of cancer, involving Nrf2 activation [66]. An important limitation in taking drugs with proven anti-inflammatory and antioxidant effects is their limited therapeutic range and safety profile. For example, corticosteroids are established as first-line drugs for the treatment of acute demyelinating attacks/exacerbations of MS [17,143,179,180]. Antioxidant effects have been established in MS patients, such as increased SOD activity and decreased ROS and MDA levels after methylprednisolone treatment [47,181,182]. However, if patients do not respond to corticosteroid therapy, which occurs in 20–25% of MS cases, a second corticosteroid pulse therapy in combination with TPE is recommended after an interval of 10–14 days [169,180,183], or switching to TPE entirely when the adverse effects of corticosteroids are presented. The latter would have its clinical rationale in terms of affecting OS as well. The same applies to DMF in the event of progressive multifocal leukoencephalopathy, severe lymphocytopenia, or a faster depletion of its antioxidant effect, as is the case in elderly patients [112,123,124,125,126].

TPE is an effective and relatively safe method that can be applied in the treatment of neurodegenerative diseases [17,184]. However, the use of TPE in the management of acute relapsing MS is still modest [184] and also has inherent limitations in terms of local specificities related to the population under investigation, experience, availability, and insurance coverage [17,165].

The most important limitations and advantages of the various Nrf2 activators are summarized in Table 2 [10,17,112,123,124,125,150,151,152,153,165,185,186].

Nrf2 is reported to either suppress or induce inflammation and immunity in a cell-type- and disease-dependent manner [69]. Nrf2 activation alters the balance between Th1/Th2 in preclinical disease models by altering Th1-driven responses and shifts T helper lymphocytes toward Th2 differentiation. On the other hand, the increased production of Nrf2 in macrophages downregulates the expression of inflammatory genes through inhibition of the NF-kB signaling pathway [69]. TPE can alter the immunological response of cells as well, however, only by shifting the Th1/Th2 balance toward Th-1 predominance [159] along with a decrease in OS and pro-inflammatory processes [143]. To what extent the combined approach of the administration of exogenous activators, together with the application of TPE and the resulting endogenous stimulation of Nrf2, may contribute to a better clinical response in MS will be the subject of future profound investigations.

In conclusion, our preliminary findings from the shared clinical experience with nanomembrane-based TPEs outline new perspectives in OS modulation. The increased plasma levels of NGF in response to TPE treatment of MS patients suggest their antioxidant potential for endogenous Nrf2 enhancement via NGF/TrkA/PI3K/Akt and NGF/p75NTR/ceramide-PKCζ/CK2 signaling pathways.

## Figures and Tables

**Figure 1 ijms-24-17223-f001:**
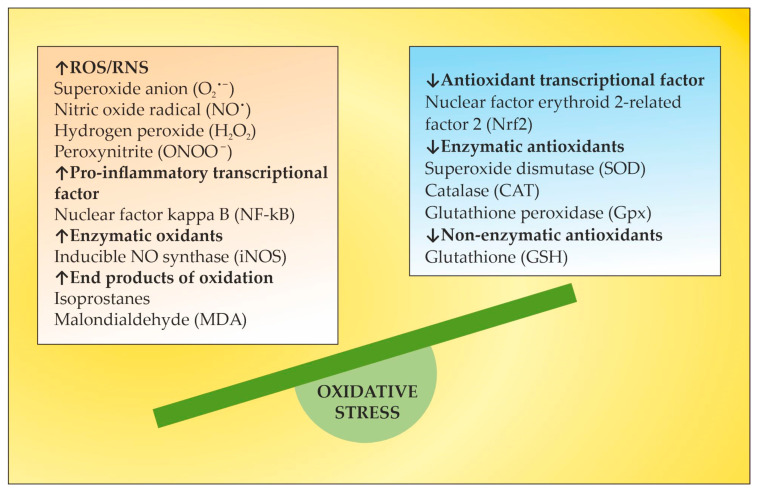
Major OS markers and factors of redox imbalance implicated in MS pathogenesis. Adapted from [22] and modified.

**Figure 2 ijms-24-17223-f002:**
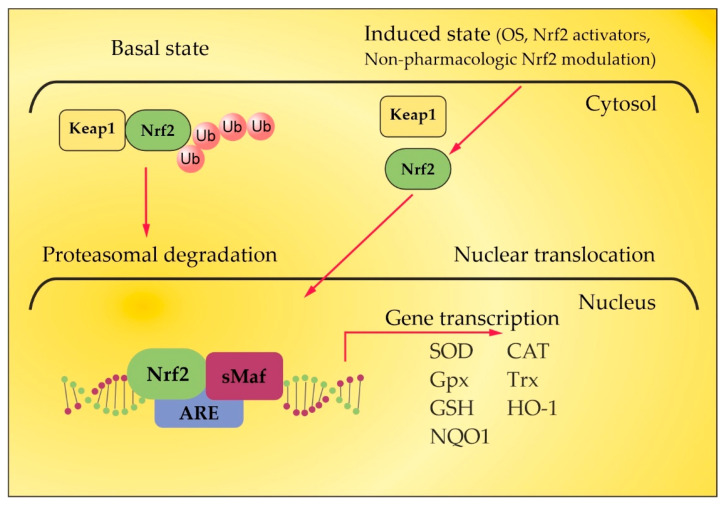
Keap1/Nrf2/ARE signaling pathway. Adapted from [52] and modified.

**Figure 3 ijms-24-17223-f003:**
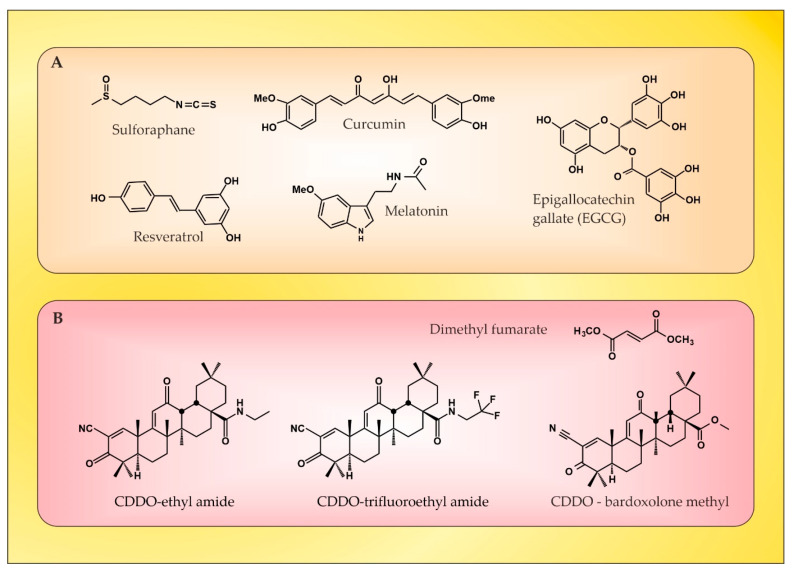
Chemical structures of certain natural (**A**) and synthetic (**B**) Nrf2 activators. Adapted from [10] and modified.

**Figure 4 ijms-24-17223-f004:**
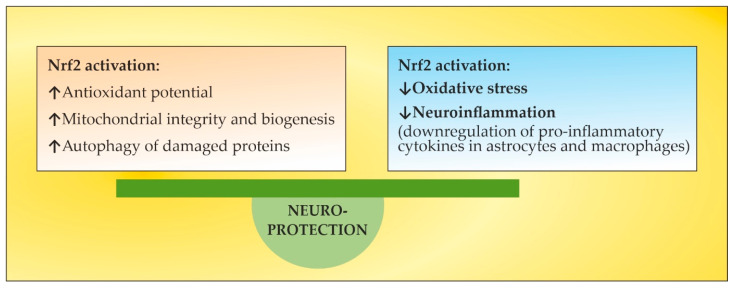
Neuroprotective effects of Nrf2 activation. Adapted from [10] and modified.

**Figure 5 ijms-24-17223-f005:**
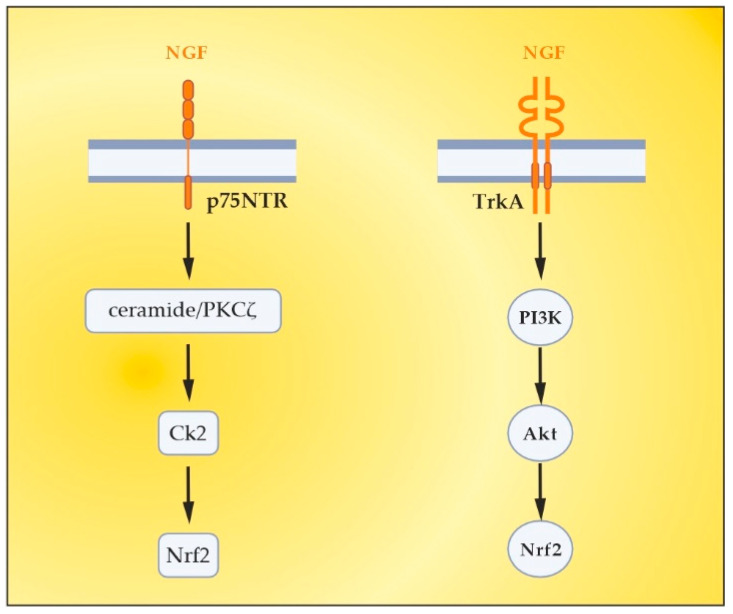
NGF-mediated activation of Nrf2 transcriptional factor. Adapted from [143] and modified.

**Figure 6 ijms-24-17223-f006:**
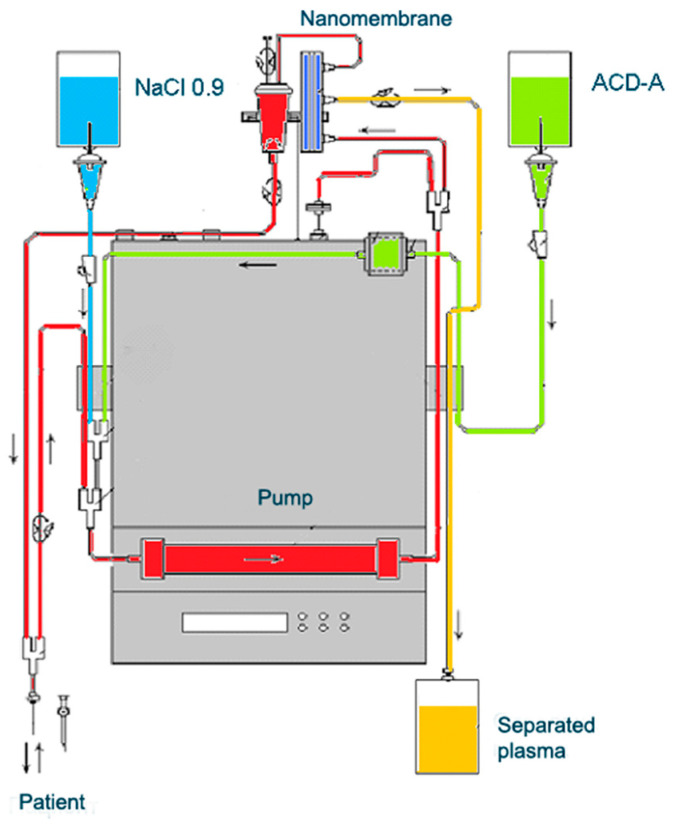
Diagram of nanomembrane-based TPE device.

**Figure 7 ijms-24-17223-f007:**
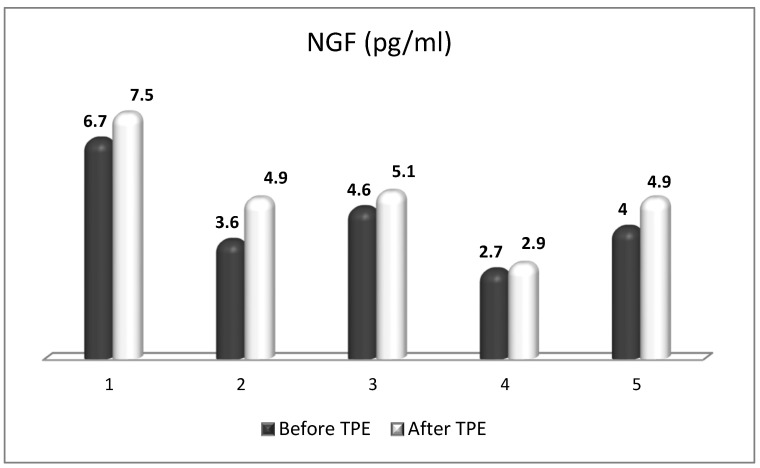
Changes in the level of NGF of patients (1, 2, 3, 4, 5) with MS before and after TPE.

**Figure 8 ijms-24-17223-f008:**
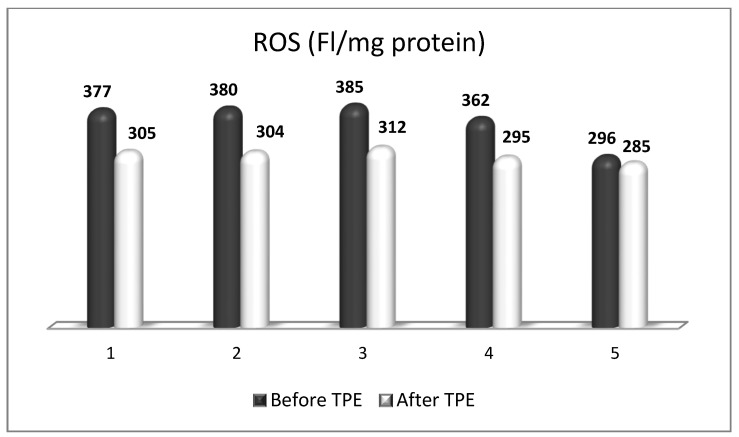
Changes in the ROS content of patients (1, 2, 3, 4, 5) with MS before and after TPE. (FI/mg protein = fluorescence intensity/mg protein).

**Figure 9 ijms-24-17223-f009:**
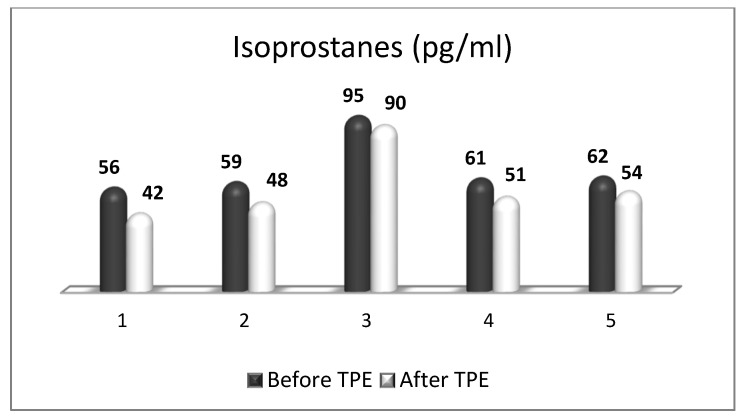
Changes in the level of isoprostanes of patients (1, 2, 3, 4, 5) with MS before and after TPE (pg/mL).

**Figure 10 ijms-24-17223-f010:**
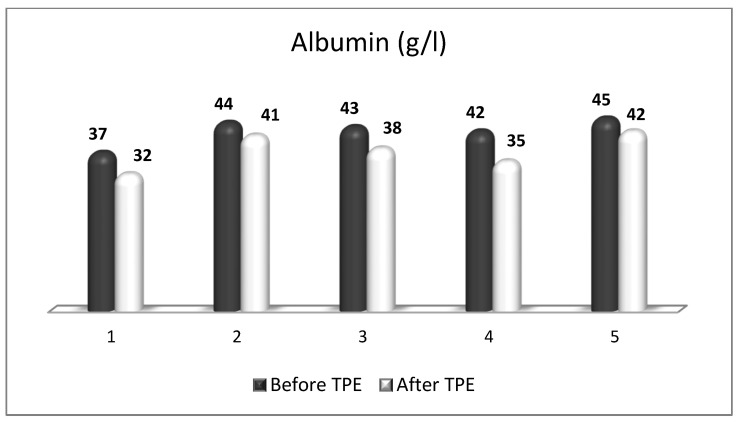
Changes in the level of albumin (normal range 35–50) of patients (1, 2, 3, 4, 5) with MS before and after TPE (g/l).

**Table 1 ijms-24-17223-t001:** The effect of plasmapheresis on OS (ROS, isoprostanes) and disability scores (Kurtzke EDSS) in patients with MS (ROS—reactive oxygen species; EDSS—expanded disability status scale).

Patient Number	ROS (FI/mg Protein)		Isoprostanes (pg/mL)		Kurtzke EDSS	
	Before TPE	After TPE	Before TPE	After TPE	Before TPE	After TPE
1.	377	305	56	42	6.5	6.0
2.	380	304	59	48	6.5	6.0
3.	385	312	95	90	8.5	8.5
4.	362	295	61	51	3.0	2.5
5.	296	285	62	54	6.0	5.5

**Table 2 ijms-24-17223-t002:** Limitations and advantages of different Nrf2 activators.

Nrf2 Activators	Limitations	Advantages
Natural exogenous	Poor drug solubility Low oral bioavailability Increased first-pass metabolism Quick biotransformation and elimination Low plasma concentrations Non-specific effects (can act on other signaling pathways, especially at high doses)	Ingestion with food Available as dietary supplements Affordable due to fair price Increasing clinical trial experience for the treatment of neurodegenerative and neuro-inflammatory CNS disorders (including MS)
Synthetic exogenous	Lymphocytopenia, leukoencephalopathy	Clinically approved in MS treatment
Endogenous	Experimental preclinical evidence Preliminary investigational clinical evidence Less affordable due to high price of TPE	No need for pharmacokinetic (drug delivery) optimization Antioxidant, anti-inflammatory, and immunomodulatory rapid clinical improvement in MS patients

## Data Availability

All data are available under request to corresponding author Dimitar Tonev (dgtsofia@abv.bg).

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
