# Peer review of "Oxidative Stress and the Nuclear Factor Erythroid 2-Related Factor 2 (Nrf2) Pathway in Multiple Sclerosis: Focus on Certain Exogenous and Endogenous Nrf2 Activators and Therapeutic Plasma Exchange Modulation"

_ijms, 2023, doi:10.3390/ijms242417223_

Round 1

Reviewer 1 Report

Comments and Suggestions for Authors

This review explores exogenous approaches to boost Nrf2 activation and discusses potential non-pharmacological therapeutic strategies. It also mentions examples of Nrf2 activators and therapeutic plasma exchange (TPE) modulation, focusing on the antioxidant potential of increased plasma NGF levels in response to TPE treatment for endogenous Nrf2 enhancement. Overall, the manuscript is interesting and important for the field. However, I have several suggestions for improvement. 

-       I noticed in several parts on the manuscript that there are extra spaces between words. The authors should do another review of the paper and make sure it looks uniform. 

-       Abstract: 

o   Line 21 to 25- These sentences are very confusing. I suggest improving. 

o   Line 27- There is an extra space at the beginning of the sentence. 

-       Introduction: 

o   Line 38- The sentence needs to be clarified. 

o   Line 42- Extra space. 

o   Line 54 to 57- It is unclear what the authors want to share in this part. I recommend clarifying and improving the sentence. 

-       Methodology: 

o   Line 79- The authors refer to a “local database”, which dabase? It should be described here. 

o   It would be interesting to see how many manuscripts have come out with the searches and how many were relevant for this study. 

-       OS in MS: 

o   Line 101- The authors could explain in more detail what is presented in Figure 1. It would improve the quality of the manuscript if the authors incorporated information on how the disease affects the different brain regions, which cell types are initially affected and it’s evolution with the disease progression. There is some attempt at explaining the cell type, but it is hard to follow. 

-       Nrf2 pathway in MS: 

o   Line 163 to 167- The sentences could be clearer. I suggest improving the clarity of the message to share. 

o   Line 200- The authors should discuss more the HO-1 role on MS treatment. 

-       Line 442 to 445- I do not understand why there are parentheses in the middle of the sentence. I recommend improving the sentences.  

-       The authors introduce results in this review. I could not comprehend if these results were new or coming from other studies. If they are described from other studies, they should be more clearly incorporated into the text. If they are new results, the methodology behind it should be described in the methodology section, including the ethical permits to work with human samples. 

Comments on the Quality of English Language

- The authors should work to improve the clarity of several sentences. 

- The authors should also improve grammar and other manuscript editing (like the extra spaces). 

Reviewer 2 Report

Comments and Suggestions for Authors

Dimitar Tonev and Albena Momchilova in this manuscript reviewed the roles of oxidative stress (OS) in the pathogenesis of multiple sclerosis (MS), focusing on the exogenous and endogenous Nrf2 activators and therapeutic plasma exchange (TPE) modulation to treat MS. The manuscript nicely provided an overall picture of ROS, NRF2 activators, and TPE in the pathogenesis and therapy of MS. But the causal or consequence effects of OS in the MS pathogenesis is too brief, the authors may want to discuss more. More importantly, the authors focused on the exogenous and endogenous Nrf2 activators and therapeutic plasma exchange (TPE) , the should add a diagram or table to compare the advantages and disadvantages of those methods for the potential MS therapy.

Minor comments:

1.      Line 47, the authors should briefly describe the concept/definition of OS for the broad scope of readers, as referenced in recent other papers, e.g. PMID: 36830722 and PMID: 32352946.

2.     Line 108, oxidative stress should be OS - the abbreviation form.

3.      Line 187-192. Besides the competitive features of NRF2 and NF-kB, NRF2 is transcriptionally upregulated by NF-kB, which acts as a feedback inhibition of NF-kB. Please see the PMID: 32640524 and PMID: 26919428.  In addition, recent studies showed that both NRF2 and ATF4 regulate OS (PMID: 18458112 and PMID: 36996941), in most cases, NRF2 and ATF4 coordinate to regulate OS. That should be discussed here.

4. In lines 379-415, the authors should provide a graph to show readers how TPE works.

Comments on the Quality of English Language

None

Round 2

Reviewer 1 Report

Comments and Suggestions for Authors

The authors have addressed my comments, and the manuscript has improved significantly. I recommend the paper for publication. 

Reviewer 2 Report

Comments and Suggestions for Authors

All concerns are addressed.

Comments on the Quality of English Language

English is OK now.